# Regulation and Functions of Protumoral Unconventional T Cells in Solid Tumors

**DOI:** 10.3390/cancers13143578

**Published:** 2021-07-16

**Authors:** Emilie Barsac, Carolina de Amat Herbozo, Loïc Gonzalez, Thomas Baranek, Thierry Mallevaey, Christophe Paget

**Affiliations:** 1Centre d’Étude des Pathologies Respiratoires, UMR1100, INSERM (Institut National de la Santé et de la Recherche Médicale), Faculté de Médecine, Université de Tours, 37012 Tours, France; emilie.barsac@etu.univ-tours.fr (E.B.); loic.gonzalez@univ-tours.fr (L.G.); baranek@univ-tours.fr (T.B.); 2Department of Immunology, University of Toronto, Toronto, ON M5S 1A8, Canada; caro.deamat@mail.utoronto.ca (C.d.A.H.); thierry.mallevaey@utoronto.ca (T.M.); 3Institute of Biomedical Engineering, University of Toronto, Toronto, ON M5S 3G9, Canada

**Keywords:** unconventional T cells, γδT, NKT, MAIT, immunoregulation, cytokines, solid cancer, protumor, tumor microenvironment

## Abstract

**Simple Summary:**

While the biology of unconventional T cells has been extensively studied in the context of infection, their role in tumor immunity has only recently emerged. In this review, we provide an overview of the current knowledge pointing towards protumoral functions of unconventional T cells in solid cancer. The tumor microenvironment factors that shape and control these deleterious properties are also reviewed. Finally, we discuss how these elements may be considered as future targets in cancer immunotherapy and the outstanding questions in the field.

**Abstract:**

The vast majority of studies on T cell biology in tumor immunity have focused on peptide-reactive conventional T cells that are restricted to polymorphic major histocompatibility complex molecules. However, emerging evidence indicated that unconventional T cells, including γδ T cells, natural killer T (NKT) cells and mucosal-associated invariant T (MAIT) cells are also involved in tumor immunity. Unconventional T cells span the innate–adaptive continuum and possess the unique ability to rapidly react to nonpeptide antigens via their conserved T cell receptors (TCRs) and/or to activating cytokines to orchestrate many aspects of the immune response. Since unconventional T cell lineages comprise discrete functional subsets, they can mediate both anti- and protumoral activities. Here, we review the current understanding of the functions and regulatory mechanisms of protumoral unconventional T cell subsets in the tumor environment. We also discuss the therapeutic potential of these deleterious subsets in solid cancers and why further feasibility studies are warranted.

## 1. Introduction

Despite tremendous advances in treatment and earlier detection, cancer remains one of the most devastating diseases worldwide, with around 20 million new cases and 10 million cancer-related deaths in 2020. It is now well established that immune cell composition within tumors has prognosis significance in a wide array of cancers. Among tumor-infiltrating leukocytes are unconventional T (UT) cells, a heterogeneous family of thymus-derived T cells that respond to antigens (Ags) that are invisible to conventional major histocompatibility complex (MHC) class I- and II-restricted CD4^+^ and CD8^+^ T cells. UT cells comprise three main lineages, namely CD1d-restricted natural killer T (NKT), MR1-restricted mucosal-associated invariant T (MAIT) and γδT cells. In line with their quasi-monomorphic restricting elements, the T cell receptors (TCRs) of UT cells recognize Ags of limited diversity, including lipid-based Ags and small metabolites. This is of interest in tumor biology in which major metabolic changes are observed and neoAgs of glycolipid nature have been reported [1,2]. In addition, UT cells express innate receptors such as natural killer (NK) receptors (NKRs) that widen their tumor cell-recognition capacity. The involvement of UT cells in tumor immunity has emerged over the last two decades using UT cell-deficient mice in many models of carcinogenesis, transplantable tumors and genetically controlled spontaneous models. In humans, a recent meta-analysis of around 18,000 tumors identified T cells co-expressing CD161 as one of the most favorable prognostic markers [3], a phenotype reminiscent of UT cells.

Although UT cells have been mainly recognized for their antitumor functions, accumulating evidence suggests that UT cells can also exert protumoral activities. This can be explained by the fact that UT cells are composed of discrete effector subsets that fit into core regulatory (T_H_1-, T_H_2- and T_H_17-like) “immune modules”, similar to conventional CD4^+^ T cells and innate lymphoid cells (ILC) [4].

In this review, we first broadly introduce fundamental UT cell biology in mice and humans. In the context of tumor recognition, we then outline UT cell activation mechanisms and functions that promote tumor progression/escape. Finally, we discuss the possible future implications of these concepts for cell-based treatments of human solid cancers.

## 2. Generalities on UT Cells

One of the most notable differences between NKT, MAIT and γδT cells is the nature of the Ags they recognize through their TCRs, namely glycolipids, riboflavin derivatives and phosphorylated metabolites of the isoprenoid pathway, respectively. The presentation and/or recognition of such ligands involves the highly conserved molecules CD1d, MR1 and butyrophilin, respectively (Table 1). Beyond this, UT cells share some developmental trajectories, activation mechanisms and effector functions.

### 2.1. UT Cell Ontogeny

UT cells emerge at different branching points during thymocyte development (Figure 1), upon rearrangement of the appropriate TCR gene segments. At the CD4^−^CD8^−^ double negative 2/3 (DN2/3) stage, when the *Tcrb*, *Tcrg* and *Tcrd* gene loci rearrange, thymocytes that express a productive γδ TCR undergo a single positive selection process and subsequent maturation to form the γδT cell lineage [26,27,28]. Discrete mouse and human γδT cell populations develop in a series of overlapping waves starting from the fetal period. These populations are defined by their TCR variable region usage, preferential homing to particular peripheral tissues and discrete effector functions (see [5,10,29] for reviews).

Both NKT and MAIT cells emerge from the CD4^+^CD8^+^ double-positive (DP) thymocyte pool, when developing thymocytes rearrange and express the TCRα chain and undergo positive selection (see [29,30] for reviews). At this stage, thymocytes expressing the canonical TRAV11-TRAJ18 (TRAV10-TRAJ18 in humans) or TRAV1.2-TRAJ33 TCRα chain and undergo positive selection will be driven to the invariant (i)NKT and MAIT cell lineages, respectively [8,31]. Unlike conventional T cells, iNKT and MAIT cells are positively selected by CD1d- and MR1-expressing DP thymocytes, respectively [8,12,31,32,33], and their development also requires SLAM family receptor activation, especially Slamf6 (Ly108) [34,35,36], and strong TCR signaling, which induces the expression of the transcription factor promyelocytic leukemia zinc finger (PLZF), a master regulator of UT cell effector programs [37,38,39,40,41,42]. Of note, a population of CD1d-restricted variable (v)NKT cells is found in mice and humans. vNKT cells respond to glycolipid Ags not recognized by iNKT cells, such as sulfatides (Table 1). Little is known about the development of these cells and their functions in disease, due to the lack of specific reagents to track them [43].

### 2.2. UT Cell Effector Differentiation

Like conventional CD4^+^ T cells and ILCs, UT cells differentiate into discrete type 1 (IFN-γ-producing), type 2 (IL-4-producing) and type 3 (IL-17-producing) effector subsets (Figure 1) that require the expression of the signature transcription factors T-bet, PLZF/GATA-3 and RORγt, respectively [36,44,45]. In mice, iNKT/MAIT/γδT1 and 17 can be found in various proportions in most tissues. Bona fide MAIT2 and γδT2 cells are virtually absent in mice [32,41,42,46,47], and the existence of a fully differentiated iNKT2 effector subset has recently been challenged [44,48]. UT cell effector subsets can differentiate in the thymus or in peripheral tissues [42,43,44,47,49,50].

It is not clear whether discrete iNKT cell subsets exist in humans. However, a recent unbiased transcriptomic analysis identified different clusters of circulating iNKT cells, including immature CD4^+^ iNKT cells, and two clusters with cytotoxic or regulatory signatures [45]. Most human MAIT cells appear to co-express T-bet and RORγt [41,46]. γδT cells expressing high levels of the TCR/CD3 complex, irrespective of their TCR repertoire, express RORγt and produce T_H_17 responses [47]. Mouse IL-17-producing Vγ6Vδ1 cells similarly express high TCR levels [51].

Although only imperfectly understood, the effector differentiation of some UT cells is influenced by TCR signal strength as well as cytokines. In addition, the functional relevance of discrete UT cell effector subsets during cancer immunity has not been fully established.

### 2.3. Activation Mechanisms of UT Cells

In addition to cognate Ag detection via their TCRs, UT cells respond to stress- or pathogen-induced surface molecules and cytokines (see [7,52,53] for reviews). This broadens the capacity of UT cells to recognize and respond to infectious microbes and tumor cells.

#### 2.3.1. TCR-Dependent Signals

iNKT cells respond to glycolipid Ags presented by CD1d, such as α-galactosylceramide (α-GalCer) [17]. Related α-glycolipid Ags have been isolated from various pathogenic or commensal microbes (see [25] for review). vNKT cells respond to different glycosphingolipids and phospholipids (see [20] for review). MAIT cells have been shown to respond to riboflavin (vitamin B2) metabolic derivatives synthetized by certain bacteria, fungi and yeast, presented in the context of MR1. These include 5-(2-oxopropylideneamino)-6-D-ribitylaminouracil (5-OP-RU) and 5-(2-oxoethylideneamino)-6-D-ribitylaminouracil (5-OE-RU) [16,54]. Although self-Ags are very likely involved in the development and peripheral activation of iNKT and MAIT cells, their precise nature remains controversial or elusive [49,50].

Human Vγ9Vδ2 T cells respond to small phosphorylated metabolites (i.e., phosphoantigens (PAgs)) produced by both microbial and mammalian cells. PAg recognition by the γδTCR is dependent on butyrophilin (BTN) and butyrophilin-like (BTNL) molecules [11]. Foreign or self-derived PAgs interact with an intracellular domain of BTN3A1 to induce TCR binding [55,56]. Recently, BTN2A1 was proposed to play a critical role in the Vγ9Vδ2^+^ T cell activation in association with BTN3A1 [57]. In addition, CD1d-restricted γδT subsets that recognize lipid-based antigens such as phosphatidylethanolamine (human Vδ1^+^ and Vδ2^+^) [58], cardiolipin (various mouse Vγ subsets) [52] and sulfatide (human Vδ1^+^) [53,59] have been identified in mice and humans. Ags that activate other γδT cell subsets in humans and mice remain elusive.

#### 2.3.2. TCR-Independent Signals

UT cells constitutively express cytokine receptors and can rapidly respond to various cytokines produced by activated dendritic cells (DCs), macrophages, neutrophils, epithelial/endothelial/stromal cells or infected/tumor cells, including IL-12, IL-1β, IL-18, IL-23, IL-7, IL-15 and type I IFNs (see [52,53,60,61] for reviews). Irrespective of their lineage, type 1 effector UT cells express high levels of IL-15, IL-12 and IL-18 receptors, whereas type 3 effector subsets express IL-7, IL-1 and IL-18 receptors. In addition, UT cells also express germline-encoded activating or inhibitory NK receptors such as NKG2D, NKG2A, CD94, NK1.1/CD161, 2B4 and KLRG1 [62]. NKG2D is particularly relevant as its ligands, such as MHC-I-like molecules (MICA, MICB) and ULBP4 are upregulated by tumor cells [63]. Another classical NK receptor, FcγRIII (CD16), enables UT cells to participate in antibody-dependent cell toxicity [63]. Finally, some γδT cells display a stable expression of the natural cytotoxicity receptors such as NKp30, NKp44 and NKp46 [63]. The contribution of these receptors in MAIT and NKT cell biology remains elusive. Of note, similar expression patterns of cytokine and innate receptors have been reported for ILC subsets [64].

### 2.4. Functional Diversity of UT Cells

UT cells play a crucial role in orchestrating innate and adaptive immune responses by sharing common immunoregulatory and effector functions such as cytotoxicity and the production of various cytokines and chemokines. UT cells also seem to play important functions in tissue physiology and remodeling (see [7,52,53,65,66] for reviews).

#### 2.4.1. Cytotoxicity

Activated UT cells can trigger apoptosis in target cells through the release of secretory granules containing granzyme B, granulysin and perforin and/or by engaging death receptors such as CD95 (Fas) and tumor necrosis factor-related apoptosis-inducing ligand (TRAIL). Thus, iNKT, MAIT and γδT cells have been shown to efficiently kill pathogen-infected cells mainly through granzyme B activities (see [67,68,69,70,71] for reviews). However, even if these three lineages appear to be fully equipped to mediate cytotoxicity, it remains unclear whether bona fide cytotoxic UT subsets exist as demonstrated for conventional T (CD8^+^ T cells) and innate lymphoid cells (NK cells). This could be highly relevant in the context of cancer. Of note, the cytotoxic potential of vNKT cells is far less understood, although they seem to mediate epithelial cell cytotoxicity during intestinal inflammation [72].

iNKT and γδT cells have been extensively investigated for their ability to directly kill tumor cells (see [60,61] for reviews). Although their killing activity was initially described to rely on perforin-dependent pathways, tumor cell eradication by iNKT and γδT cells can also be mediated through ligation of TRAIL on tumor cells [73,74]. In addition, the engagement of CD95 on tumor cell lines by FasL-expressing iNKT and γδT cells can also mediate tumor cell death [65,75]. Whether MAIT cells can directly kill tumor cells remains unclear. However, MAIT cells from colorectal cancer patients presented an exhausted phenotype associated with reduced granzyme B production [66]. In addition, checkpoint blockade therapy (anti-PD1) in patients with metastatic melanoma enhanced granzyme B production by MAIT cells [76].

#### 2.4.2. Release of Immunoregulatory Factors

Like conventional CD4^+^ T cells and ILCs, activated UT cell subsets can selectively produce copious amounts of cytokines related to T_H_1 (IFN-γ, TNF-α), T_H_2 (IL-4, IL-5, IL-13) or T_H_17/22 (IL-17, IL-21, IL-22, GM-CSF) (see [52,53,77] for reviews). Thus, they can modulate the activation status of several other leukocytes such as neutrophils, macrophages, NK cells, DCs, regulatory T cells, conventional CD4^+^ and CD8^+^ T cells and B cells (see [7,53,78] for reviews). Cross-regulation has also been reported between UT cells from different (sub)lineages [77,79,80,81], although the interplay between MAIT cells and other UT cells remains to be demonstrated. Another way for UT cells to interact with immune cells is through the release of CC and CXC chemokines such as CCL3, CCL4, CCL5, CCL11 and CXCL10 enabling them to recruit NK cells, macrophages, neutrophils, eosinophils, DCs and conventional T cells (see [52,53,77] for reviews).

### 2.5. UT Cell Populations: Redundant Functions for Specific Roles?

The multilayered organization of the immune system raises the issue of uselessness or redundancy in certain cellular compartments [82]. According to their quite conserved functions, such a question might apply to UT cells. It is, however, important to mention that these three lineages have been co-conserved in mammals [83] with few exceptions (e.g., lagomorphs and cattle). Besides their non-redundant antigenic repertoire, UT cell effector subsets seem to seed lymphoid and nonlymphoid tissues with a specific positioning. For instance, a recent study reported that MAIT and iNKT cells populate the same peripheral tissues but with inverse proportions of T_H_1-like and T_H_17-like subsets [84]. A specific role for UT cell populations in tissue physiology has been recently unraveled (see [67,68,69] for reviews). Thus, γδT cells play an important role in would healing in the skin [85] and inhibition of fibrosis in the lungs [86]. In adipose tissue in which they are abundant, UT cells seem to regulate thermogenesis, tissue innervation, insulin resistance and lipid metabolism [67,68,69]. However, some controversies have been reported, and functional investigation of the three populations at once should be encouraged. γδT cells have been recently been shown to infiltrate immune-privileged sites such as meninges and testis [78,87,88,89]. The potential presence and biological activity of iNKT and MAIT cells in these tissues have not been investigated yet.

## 3. Influence of the Tumor Microenvironment on UT Cell Functions

The tumor microenvironment (TME) is a complex milieu orchestrated by the tumor and comprising a variety of molecular and cellular determinants. In this section, we review the influence of tumor-derived ligands, TME soluble factors and microbiota on intratumoral UT cell biology (e.g., recruitment, polarization, activation, exhaustion) (Figure 2).

### 3.1. Tumor-Derived Antigens

The poor immunogenicity of classical MHC class I- and II-restricted tumor antigens is multifactorial including low expression of MHC molecules, antigenic discontinuum and instability of the peptide–MHC complex [90]. The immunogenic potential and biological relevance of tumor-derived UT cell ligands are less understood. However, tumor cells display an active metabolic flux leading to intense regulation of particular pathways [91,92] that could uncover UT cell Ags.

For instance, mouse and human iNKT cells have been shown to react to CD1d-expressing tumor cells [93,94], suggesting that tumor cells can generate/uncover and present CD1d-dependent iNKT cell Ags. Gangliosides such as the monosialylganglioside GM3 and disialylgangliosides GD3 and GD2 as well as modified species have been recognized as tumor Ags in multiple cancer types [95]. Tumor-derived GM3 (N-glycolyl-GM3) and GD3 have been shown to regulate iNKT cell activity, including proliferation and cytokine profile, in a CD1d-dependent manner [96,97,98]. Using bulk gangliosides, other studies have reported an inhibitory potential for GM3 and GD3 on TCR-mediated iNKT cell activation [99,100,101]. We recently demonstrated that GM3 and GD3 comprising a d18:1–C24:1 ceramide backbone were able to activate iNKT cells while other species failed to do so [24]. This may partially explain the conflicting results since bulk gangliosides contain various ceramide species resulting in a potential competition between inhibiting and activating molecules. For example, C24:1 gangliosides are almost absent in commercial bovine buttermilk-derived GM3 [24], a common source of gangliosides for bioassays. Although it is unclear whether iNKT cells are able to react to the GD2 ganglioside in a natural setting, human iNKT cells engineered to express a chimeric antigen receptor against GD2 are a promising novel therapy to treat neuroblastoma [102,103,104]. Interestingly, ER stress, which is activated in tumor cells through the unfolded protein response [105], has been recently shown to generate endogenous neutral ligands that could activate iNKT cells [106].

The ability of vNKT cells to recognize tumor Ags has not been described so far. It is interesting to note that most of the identified CD1d-restricted vNKT ligands are phospholipids (e.g., lysophosphatidylcholine, phosphatidylethanolamine, phosphatidylglycerol and cardiolipin) [107], which are highly enriched in mitochondrial membranes [108]. The mitochondrial reprogramming that occurs during tumorigenesis and cancer progression [109] might therefore affect phospholipid metabolism to generate CD1d-dependent vNKT cell regulators. To date, the best characterized Ag for vNKT cells is the glycosphingolipid C24:1 sulfatide [110]. Sulfatides are commonly found in mouse and human tumor cells. Although C24:1 sulfatide is one of the major species under physiological conditions, it appears to constitute a minor fraction in transformed cells [111].

As mentioned previously, tumor cells can express high levels of some PAgs, such as isopentyl pyrophosphate (IPP), which activates the Vγ9Vδ2 T cells [112]. IPP accumulation in tumor cells is a complex and not entirely understood process but might rely on the reduced expression of the IPP-consuming enzyme, farnesyl pyrophosphate synthase [92]. BTN molecules are critical in the PAg-dependent activation of Vγ9Vδ2 T cells [55,56,57] by tumor cells, although a mechanism involving a cell surface F1-ATPase-like/apolipoprotein A1 complex has also been proposed [113].

Mammalian cells, including transformed cells, are believed to lack a functional riboflavin pathway, precluding the generation of the currently identified MR1-restricted MAIT cell ligands. However, a recent study demonstrated a key MR1–MAIT cell axis controlling tumor growth and metastasis [114]. Defining the source and identity of the “endogenous” ligands in this system is an outstanding question. Of note, tumor cell-derived MR1-restricted Ags have been recently identified but were shown to activate non-MAIT MR1-restricted T cells [115].

### 3.2. Cytokines, Metabolites, pH and Hypoxia

Many soluble factors, such as cytokines, metabolites, nutrients or oxygen, can influence tumor-infiltrating UT cells to foster a local immunosuppressive environment, which promotes tumor growth.

Several cytokines such as IL-1β, IL-6, IL-7, IL-23 and TGF-β produced within the TME are involved in polarization, activation and proliferation of protumoral γδT cells associated with a T_H_17-like profile. In breast cancer and Lewis lung carcinoma metastatic model, IL-1β induces IL-17 production by tumor-infiltrating γδT cells [116,117]. In fibrosarcoma, Wakita and colleagues showed that IL-17 in the TME was mostly produced by γδT cells. In this study, IL-6 and TGF-β polarized intratumoral γδT cells towards a γδT17 profile, which produced IL-17 in an IL-23-dependent manner [118]. Elevated expression of IL-7 is associated with poor prognosis in various cancer types [119,120,121]. Furthermore, tumor-derived IL-7 controls the accumulation and activity of both protumoral γδT17 and iNKT17 cells in breast and ovarian cancer models [122,123]. We demonstrated a critical role for type I IFN signaling in negatively controlling IL-7-dependent accumulation of γδT17 cells in the tumor bed [122]. Since we and others found that IL-7 was a preferential proliferative factor for the three lineages of IL-17-producing UT cells [124,125,126,127], targeting the IL-7/UT17 cell axis might be of interest for therapeutic purposes.

In addition to promoting a T_H_17-like profile, TGF-β was shown to convert Vγ9Vδ2 T cells with a regulatory profile characterized by the expression of Foxp3. These cells expressed inhibitory receptors such as ICOS, CD25 and GITR and secreted high levels of TGF-β and IL-10 capable of inhibiting T cell proliferation and activation [128,129]. Similarly, the addition of TGF-β to TCR-activated iNKT cells induces suppressive Foxp3^+^ iNKT cells in humans and mice [130,131]. Interestingly, in human colorectal cancer, TGF-β induced tumor-infiltrating regulatory γδT cells expressing high levels of CD39, CTLA-4, CD25, PD-1 and Foxp3 [132]. Upon stimulation, this subset produced IL-10, IL-17 and GM-CSF [132]. A similar exhausted phenotype was reported for MAIT cells from colon cancer patients [66], a cancer type in which TGF-β is produced in high levels [133].

IL-4 is the prominent T_H_2-related cytokine and has been associated with the proliferation and survival of several cancer cells. The addition of IL-4 to PBMC from healthy donors promoted the growth of IL-10-producing Vδ1 T cells [134]. In B16F10 melanoma model, genetic or antibody-mediated IL-4 blockade promoted Vγ4 T cell-dependent antitumor functions [135]. The IL-4 receptor-signaling also induces IL-13 production by vNKT cells, resulting in the repression of tumor immunosurveillance in a model of fibrosarcoma [136]. Importantly, this phenomenon was not found in IL-4-deficient animals, suggesting an autocrine role for IL-13 through IL-4R signaling.

Although IL-21 has been mainly described to enhance cytotoxic functions of both CD8^+^ T and NK cells [137], it has recently been associated with oncogenic properties [138]. Similar to the phenotype obtained under TGF-β pressure, expansion of Vγ9Vδ2 or Vδ1 T cells in presence of IL-21 promoted the emergence of a regulatory subtype capable of producing IL-10 [139,140]. In addition, rIL-21 administration in patients with stage IV metastatic melanoma reduced IFN-γ and TNF-α production and enhanced the production of IL-4 by iNKT cells [141]. These data may support a regulatory effect of IL-21 on the protumoral functions of iNKT cells.

Tumors grow in an altered environment characterized by an acidic extracellular milieu, dysregulated metabolic pathways and hypoxia [142,143,144]. Recently, two studies showed that lactic acid in the TME impairs the metabolism and IFN-γ production of iNKT cells in mice and humans [145,146]. This effect was proposed to rely on inhibition of mammalian target of rapamycin (mTOR) signaling and nuclear translocation of PLZF [145]. Interestingly, a recent study showed defective mTORC1-dependent glycolytic metabolism and type 1 effector response in MAIT cells of obese patients [147] that may have some implications in cancer [148].

The arachidonic acid metabolite prostaglandin E_2_ (PGE2) derives from cyclooxygenase-2 (COX-2) and is associated with poor clinical outcome in various cancers. PGE2 production by mesenchymal stem cells was described to suppress antitumoral Vγ9Vδ2 T cell response through the engagement of E-prostanoid 2 (EP2) and EP4 receptors [149,150]. In addition, γδT cell-mediated lysis was inhibited by increased release of PGE2 by pancreatic tumor cells linked to an enhanced COX-2 expression [151]. PGE2 produced by human lung cancer cells was also proposed to inhibit the secretion of granzyme B, perforin and IFN-γ by CD3^+^ CD56^+^ cells, which are reminiscent of UT cells [152]. Finally, COX-2 inhibition restored the proliferative capacity of blood iNKT cells isolated from patients with laryngeal cancer [153]. Recently, palmitic acid and cholesterol have been described as key promoting factors for γδT17 cells [154]. Cholesterol treatment boosted γδT17 cell proliferation and high-fat diet promoted their expansion and accumulation in different tumor models. Interestingly, lipase inhibition induced a decreased number of intratumoral γδT17 cells associated with a reduced growth of B16F10 tumors in mice [154]. Palmitic acid also induces iNKT cell protumoral functions by inducing the degradation of *Tbx21* transcripts and therefore blunting IFN-γ production [155].

Hypoxia is an important factor in the host antitumor response. Recently, Park and colleagues elegantly showed that antitumoral functions of brain tumor-infiltrating γδT cells were impaired through HIF-1α-induced apoptosis and NKG2D downregulation. Interestingly, reoxygenation of the TME restored NKG2D-related antitumoral functions of γδT cells [156].

### 3.3. Microbiota

The microbiota contributes to the proper development, maturation and function of the immune system, thereby influencing homeostatic immunity and disease susceptibility. Commensal microbes have been shown to regulate tumor immunosurveillance and, more recently, responses to checkpoint blockade cancer immunotherapies [157,158]. Germ-free (GF) mice, which are devoid of commensal microbes, have multiple and diverse defects in UT cell homeostasis and function. Most strikingly, MAIT cells are virtually absent in GF mice, and although they still develop in the thymus of these animals, residual MAIT cells fail to proliferate and mature properly [12,42,159,160]. γδT and iNKT cell defects in GF mice are more subtle, tissue-specific and relate more to their effector differentiation and/or functional response. A seminal study from Lefrancois and Goodman demonstrated that intestinal γδT cells in GF mice lack cytotoxic activity [161]. In addition, γδT17 cells are drastically reduced in GF mice, including in the intestines, lungs and the skin [58,162,163], and intestinal γδT17 cells in GF mice fail to respond to bacterial infection [164]. The development and tissue distribution of iNKT cells are largely unaffected in GF mice, except for an increased prevalence in the lungs and intestines [165,166].

Normal UT cell homeostasis can be restored through the colonization of GF mice with a complex microbiota or selected microbes. Interestingly, early life microbial colonization of GF mice, but not post-weaning, restores proper iNKT and MAIT cell homeostasis and function [160,167,168]. Normal MAIT numbers and functions are restored in GF mice by the colonization with riboflavin-expressing bacteria such as *Bacteroidetes thetaiotaomicron*, *Lactobacillus casei* or *Escherichia coli* but not by bacteria deficient in the riboflavin synthesis pathway such as *Enterococcus faecalis* [160,168,169]. Similarly, colonization with *Bacteroidetes fragilis* or *Sphingomonas yanoikuyake*, which express iNKT cell ligands, but not *E. coli*, can restore proper iNKT cell homeostasis [165,170]. Finally, the skin commensals Corynebacteria and their cell wall component mycolic acid promote the accumulation of skin-resident γδT17 cells [171].

The role that commensal microorganisms—or their by-products—play in fine-tuning UT cell responses and their ability to regulate antitumor responses is emerging. A seminal study by Jin et al. demonstrated that the microbiota promotes the accumulation of IL-17-producing Vγ6Vδ1 cells in the lungs and lymphoid tissues, which fuels inflammation and promotes lung adenocarcinoma induced by Kras mutation and p53 loss [172]. The microbiota appears to impact iNKT cell homeostasis and response during lung and gut inflammation [166,173]. However, it is unclear whether iNKT cells themselves were altered in these studies. Interestingly, dysbiosis induced by antibiotic treatment leads to the accumulation of IFN-γ-producing iNKT cells (presumably iNKT1) in the liver, which prevents tissue regeneration following partial hepatectomy but provides antitumor surveillance [159,174]. MAIT cell development critically relies on the microbiota. However, whether dysbiosis (e.g., antibiotic treatment) affects their responses is currently unknown.

## 4. Emerging Protumoral Functions of Intratumoral Unconventional T Cells

Both IL-17 and GM-CSF have been associated with protumoral activities in experimental models of solid cancers resulting in increased tumor growth and metastasis. Recent evidence has indicated that UT cells represent an early and prime source of these two mediators within the tumor microenvironment. The protumoral activities of UT cells are largely linked—but not restricted—to their ability to secrete IL-17 and GM-CSF. The mechanisms underlying protumoral UT cell activities involve different pathways such as increased angiogenesis, recruitment of immune cells with suppressive activities and direct inhibition of antitumor functions (Figure 3).

### 4.1. Angiogenesis and Tumor Cell Proliferation

Angiogenesis is a key protumoral physiological process that controls tumor outgrowth and metastasis through de novo formation of blood vessels, which provide oxygen and nutrients to tumor cells, as well as filtrating metabolic byproducts, therefore fighting against hypoxia and cell death. IL-17 [160] and GM-CSF [175] have been shown to control the expression of proangiogenic factors by tumor cells, including vascular endothelial growth factors (VEGF), angiopoietins and TGF-β. As important IL-17 producers, γδT cells have been associated with the synthesis of proangiogenic factors in models of fibrosarcomas and ovarian carcinomas [118,123,162,163,176]. γδT cell-dependent effects of IL-17 on the endothelium could also control vessel permeability, favoring metastasis formation in the lung tissue [177]. Since intratumoral iNKT and MAIT cells have been shown to produce IL-17 and/or GM-CSF in experimental and clinical studies [122,167], they might also play a part in the angiogenic process. Protumoral UT cells could also directly promote angiogenesis by producing classical proangiogenic soluble factors such as TNF-α and angiopoietin-2 [168]. For instance, transcriptional analysis of tumor-associated MAIT cells in hepatocellular carcinoma indicated upregulation of *ANGPT2* (encoding for angiopoietin-2) that may suggest proangiogenic functions [169]. Recent studies showed that both murine and human MAIT cells can exhibit a transcriptomic profile associated with angiogenic properties following TCR engagement [178,179]. Although it remains unclear whether TCR-dependent activation of MAIT cells can occur in the TME, a recent study suggested a MR1-dependent MAIT activation in murine models of cancer [114].

Protumoral UT cells may promote tumor cell proliferation. For example, tumor-infiltrating MAIT and γδT cells secrete CXCL8/IL-8 [139,169], a chemokine that promotes tumor cell proliferation [180]. Moreover, γδT cells can promote lung tumor proliferation through the release of IL-22 and amphiregulin [172,181]. It is noteworthy that MAIT and iNKT cells have been shown to produce these mediators in particular contexts [182,183,184].

### 4.2. Shaping the Immunosuppressive TME

An immunosuppressive environment is a milestone in tumor progression and metastasis [185]. Within the TME, suppressive leukocytes comprise myeloid cells such as tumor-associated macrophages, tumor-associated neutrophils, tolerogenic dendritic cells (DCs) and myeloid-derived suppressor cells (MDSCs) as well as regulatory T cells (Tregs), all of them being associated with poor prognosis and chemotherapy resistance [186,187,188,189]. The protumoral activities of these subsets are primarily linked to the inhibition of the protective NK and CD8^+^ T cell response (e.g., IFN-γ production and cytotoxicity) and have been extensively reviewed elsewhere [186,187,188].

By secreting high amounts of IL-17 and GM-CSF within the TME, γδT17 cells have been shown to promote the recruitment, expansion and polarization of immune cells with suppressive functions. First, γδT17 cells were shown to contribute to the recruitment of small peritoneal macrophages and MDSCs in experimental models of ovarian [123] and lung cancers [117], respectively. In breast and lung cancer models, γδT17 cells may induce suppressive neutrophils, leading to enhanced tumor growth [172] and metastasis development [116]. Interestingly, a similar γδT17-immunosuppressive myeloid cell axis has been confirmed in humans [190]. Specifically, IL-17-producing Vδ1 T cells have been shown to recruit MDSCs into the TME in colorectal cancers [190]. Whether or not MAIT and/or iNKT cells could control similar pathways remains to be determined. Nevertheless, the frequency of MAIT cells in cervical cancer was positively associated with the presence of MDSCs [191].

vNKT cells have been shown to induce tolerogenic DCs [77], although the relevance of this phenomenon in cancer has yet to be determined. On the other hand, vNKT cells control MDSC activities (e.g., TGF-β secretion) through the release of IL-13 in models of fibrosarcoma and colon cancer [136,192]. Since they have been recently demonstrated to produce copious amounts of IL-13 upon sustained activation [193], it is possible that MAIT cells contribute to an immunosuppressive TME.

Although their functions are more associated with antitumoral immunity, iNKT cells were reported to drive Treg maintenance and activation, as well as differentiation of M2-like macrophages, in a mouse model of spontaneous polyps [194]. The ability of human iNKT cells to prime Tregs has been reported and appears to rely on the quality and intensity of iNKT cell stimulation [195].

Expression of the ectonucleotidase CD73 has been demonstrated on a substantial proportion of tumor-infiltrating human and mouse γδT cells [139,140]. CD73 regulates tumor immunity through the generation of adenosine, a metabolite with immunosuppressive functions on macrophages, T cells and NK cells [196]. Although the biological relevance needs to be demonstrated in vivo, this axis may contribute to shaping the immunosuppressive TME.

### 4.3. Inhibition of Antitumoral Functions

Additionally, protumoral UT cells can exert suppressive activities on antitumoral cells including cytotoxic T cells, NK cells and DCs.

Recently, it was suggested that MAIT cells could suppress NK cell functions in a mechanism that is dependent on IL-17 and MR1 expression by tumor cells, resulting in faster tumor growth and metastasis [114]. In addition, pulsing tumor cells with MAIT Ags resulted in increased B16F10 lung metastasis [114].

Interestingly, γδT cells can directly suppress antitumoral cells independently of their secretion of IL-17. First, human γδT cells from breast cancers can blunt DC maturation and subsequent T cell priming [182]. In addition, IL-4 production by Vγ1 T cells impairs the cytotoxic properties (e.g., NKG2D and perforin) of antitumoral Vγ4 T cells on melanoma cells [135]. The regulatory role of peculiar γδT subsets on antitumoral T cells including γδT subsets was also reported in humans [139,182]. Moreover, tumor-infiltrating γδT cells can also potently suppress CD8^+^ T cell responses against ovarian and pancreatic cancers through the release of galectin-1 [183] (an apoptotic mediator of T cells [197]) and galectin-9 [198] (a Tim-3 ligand [184]), respectively. PD-L1-expressing human γδT cells could also directly control CD4^+^ and CD8^+^ T cell infiltration in pancreatic ductal adenocarcinoma [198]. The biological relevance of this mechanism remains to be demonstrated since PD-L1-expressing myeloid cells largely outnumber their γδT cell counterparts in the TME. However, a preferential spatial γδT–αβT cell interaction within the TME could explain this effect [198].

## 5. Conclusions and Future Directions

Based on preclinical and clinical studies, UT cells have emerged as pivotal actors in cancer immunity. However, a better understanding of UT cell biology is warranted before these cells can be targeted in innovative and effective cancer immunotherapies. Many inconsistencies persist between studies according to the UT cell lineage studied and the type of malignancy considered. As discussed here, this may at least in part rely on the existence of discrete subsets endowed with opposite (anti- vs. protumoral) functions. Since UT cells can sometimes play similar functions, integrative studies investigating the various lineages of UT cells should be encouraged to better decipher their respective functions and role in a defined and unique setting. In addition, the mechanisms that drive the selective recruitment of UT cell subsets within the tumor bed remain ill-defined. How UT cells integrate within the intratumoral immune response and interact with immune and nonimmune partners should also be carefully addressed. The development of high-throughput technologies at the single-cell scale will undoubtedly enable the generation of comprehensive data to assess the heterogeneity, TCR repertoire and functional versatility of UT cells in experimental models and in cancer patients with respect to disease evolution. Such translational approaches are warranted.

Nevertheless, UT cell-based cancer immunotherapies have already been considered. To date, UT cell-based clinical trials have shown limited success, although such strategies were proven feasible and safe [23,199,200,201]. The limited positive outcome observed with UT cell-based cancer immunotherapies may be linked to the reduced proportion of UT cells in cancer patients and/or the emergence of exhausted phenotype upon repeated stimulations. These challenges may be overcome by the development of strategies to expand UT cells ex vivo with a desired functional profile (e.g., enhanced cytotoxic capacities) [202,203,204]. Irrespective of their lineages, direct and indirect protumoral functions of UT cells are mainly supported by their ability to secrete IL-17 and/or GM-CSF. Since TCR-dependent activation of UT cells activates all subsets, including those endowed with protumoral functions, the use of UT cell Ags such as α-GalCer or 5-OP-RU could also culminate in deleterious effects. In this context, protocols exploiting UT cell biology need to be revised to provide tailored treatment options for cancer patients. Thus, it is tempting to speculate that inhibition or reprogramming of these T_H_17-like UT populations might be valuable as a standalone or combined cancer treatment.

Among potential targets are the T_H_17-promoting cytokines such as IL-1β, TGF-β and IL-7. Interestingly, blockade of these cytokines or their respective receptors has shown promising clinical outcomes in several types of cancers, including leukemia (anti-IL-7Rα [205]), lung (anti-IL-1β [206]) and skin (anti-TGF-β1 [207]) cancers. Thus, the antitumoral effects of these molecules might at least partially rely on the regulation of protumoral UT cells. For instance, IL-7 is a conserved cytokine that exerts a preferential proliferative effect on all human IL-17-producing UT cell lineages [124,125,126,127], and its targeting in the context of solid tumors should be further considered.

Other opportunities could lie in reprogramming the effector functions of UT cells. Balancing IL-17 and IFN-γ production by UT cells may have therapeutic benefit, although the functional plasticity of human UT cells has been so far overlooked. Given their importance in determining UT cell functions, the manipulation of the microbiota and/or cancer metabolic pathways is attractive from a therapeutic perspective, especially for epithelial cancers [160,167,168,170]. For instance, manipulating the metabolic fitness of UT cells may be an interesting avenue of research in cancer immunotherapy. Intervention in TME metabolism may include supplementation of cancer patients with glucose, nucleic acids and/or amino acids to enhance or rewire UT cell cytotoxic functions, as recently shown for other cytotoxic lymphocytes [208,209,210,211,212]. In addition, UT cell effector functions are also controlled by post-transcriptional regulators such as epigenetic mechanisms [213] and miRNA [214,215,216]. Thus, the use of epigenetic modifiers and miRNA inhibitors is worth investigating. Recent evidence demonstrates that IL-17-producing cell functions are attuned to circadian rhythm [217] and diet [218]; therefore, these parameters should be integrated for the design of new therapeutic protocols. The design of new synthetic UT cell Ags with differential capacities to activate T_H_-related subsets should be encouraged as explored for iNKT cells. In conclusion, although a better molecular characterization of intratumoral UT cell responses is needed, UT cells undeniably represent relevant targets for future innovative antitumor strategies.

## Figures and Tables

**Figure 1 cancers-13-03578-f001:**
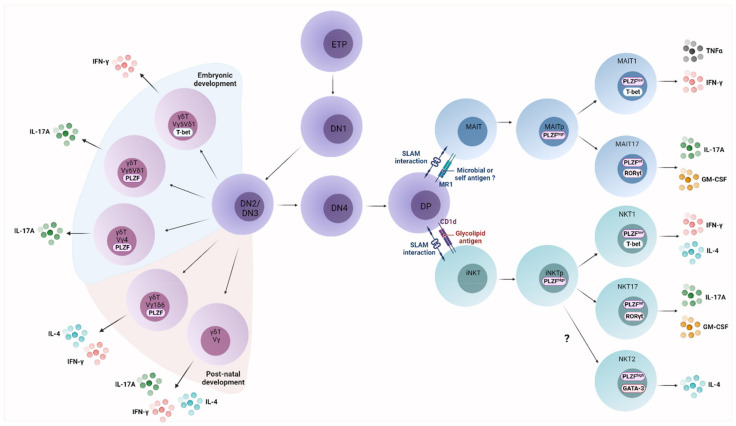
UT cell development. UT cell lineages branch away from mainstream T cells at different stages (DN2/3 for γδT cells and DP for MAIT and iNKT cells) within the thymus. TCR usage, cytokine profile and key transcription factors that define and control the differentiation of each subset are depicted.

**Figure 2 cancers-13-03578-f002:**
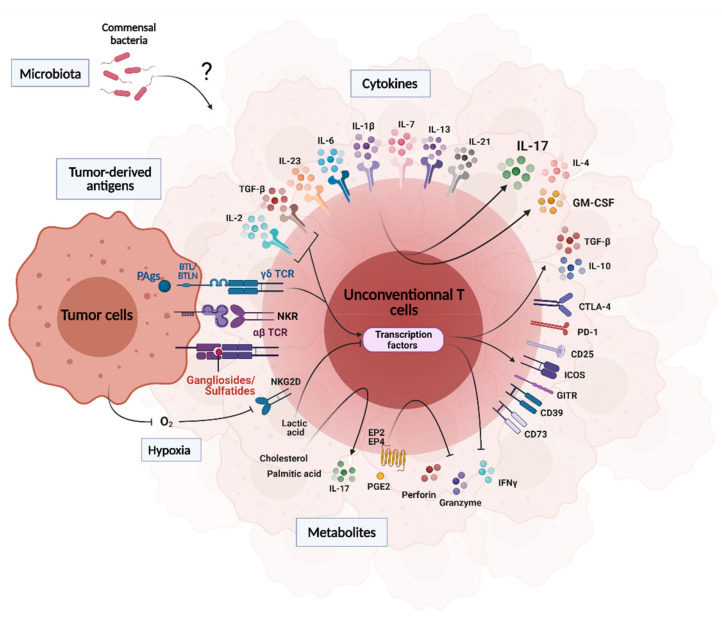
Tumor environment factors that regulate UT cell activity and functions. The regulatory processes depicted here are different from one UT cell subset to another. A close interplay occurs between the TME and UT cell biology. This is mediated by several mechanisms, including cytokine–cytokine receptor ligation, sensing of metabolites and chemical changes (O_2_ and pH). UT cells may also interact with microbiota and/or microbiota-derived factors within the TME. Finally, UT cells can react to TME-derived Ags through their TCRs.

**Figure 3 cancers-13-03578-f003:**
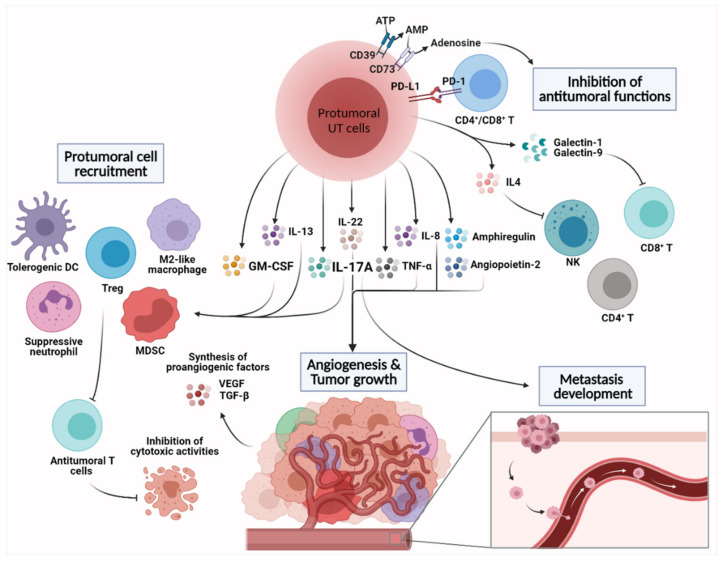
Protumoral functions of UT cells in the tumor microenvironment. The protumural functions of UT cells are mainly associated with the release of cytokines, chemokines and growth factors. In turn, these mediators have roles in the induction of angiogenesis and tumor cell proliferation. They can also contribute to the recruitment of multiple myeloid sublineages with immunosuppressive functions. Finally, UT cell-derived cytokines, especially IL-17, can trigger metastasis development. Inhibition of cells with antitumor functions is another pathway for tumor-promoting UT cells. This includes the suppression of NK and T cell responses through the production of adenosine; engagement of PD-1 by PD-L1; and the release of IL-4, galectin-1 and galectin-9.

**Table 1 cancers-13-03578-t001:** TCR repertoire, restriction and antigens of the main UT cell lineages.

Lineages		γδT Cells		MAIT Cells		iNKT Cells		vNKT Cells	
Species		*Mouse*	*Human*	*Mouse*	*Human*	*Mouse*	*Human*	*Mouse*	*Human*
**TCR repertoire**		Restricted including germline-encoded TCRs (e.g., Vγ1Vδ6.3, Vγ5Vδ1, Vγ6Vδ1) [5]	Semi-invariant or variant Restricted number of γ and δ chains [5]	Semi-invariant Vα19-Jα33 Restricted number of β chains [6,7]	Semi-invariant Vα7.2-Jα33 Restricted number of β chains [7]	Semi-invariant Vα14-Jα18 Restricted number of β chains [8]	Semi-invariant Vα24-Jα18 Restricted number of β chains [8]	Diverse, including oligoclonal Vα3.2Jα7, Vα1Jα9 Restricted number of β chains including Vβ8.1 and Vβ3.1 segments [9]	Diverse [9]
**Restricting elements**		Mainly unknown CD1d T10/T22 butyrophilin-like molecules (Skint-1) [10]	Butyrophilin 3A1/2A1 (Vγ9Vδ2) CD1d (Vδ1) CD1c viral glycoproteins [11]	MR1 [12]		CD1d [13]		CD1d [9]	
**TCR ligands**	**Natural**	Unknown Cardiolipin Phycoerythrin [14]	IPP, HMBPP (Vγ9Vδ2) Sulfatide and α-GalCer (Vδ1) EPCR (Vγ4Vδ5) [15]	Microbial-derived vitamin B2 metabolites (5-OP-RU, 5-OE-RU) [16]		α-GalCer Microbial α-derived glycolipids Ganglioside [17,18,19]		Sulfatide, LPC, PG Hydrophobic peptides β-GlcCer, LysoGL1 [20]	Sulfatide, LPC, PG β-GlcCer, LysoGL1 [20]
	**Synthetic**	-	Zoledronate *, Pamidronate * (Vγ9Vδ2) [21]	5-OP-RU derivatives [16,22]		KRN7000 ** and derivatives Ganglioside (C24:1 GM3 and GD3) [23,24]	KRN7000 ** and derivatives [25]	-	-

MAIT, mucosal-associated invariant T; iNKT, invariant natural killer T; vNKT, variant natural killer T; 5-OP-RU, 5-(2-oxopropylideneamino)-6-d-ribitylaminouracil; 5-OE-RU, 5-(2-oxoethylideneamino)-6-D-ribitylaminouracil; α-GalCer, α-galactosylceramide; β-GlcCer, β-glucosylceramide; LPC, lysophosphatidylcholine; LysoGL1, glucosylsphingosine; PG, phosphatidylglycerol. *, Aminobisphosphonates that inhibit farnesyl pyrophosphate synthase culminating in intracellular IPP accumulation; **, synthetic analog of α-galactosylceramide.

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
