# Peer review of "Regulation and Functions of Protumoral Unconventional T Cells in Solid Tumors"

_cancers, 2021, doi:10.3390/cancers13143578_

Round 1

Reviewer 1 Report

This review of unconventional T cells is well researched and has the potential to be an excellent resource for researchers in this area, but changes should be made to improve it. The main areas that need attention are:

  1. The title of the article suggests a focus on the role of unconventional T cells in cancer, but a lot of detail is provided on the development and differentiation of these cells without substantial relevance to their role in cancer immunity. As such, the article sometimes reads like a broad review of these cells instead of being focused on cancer. I suggest those sections could be condensed so they are an introduction to the main themes, rather than equally weighted. The differentiation section is more obviously relevant, but I suggest the authors provide more specific discussion about how these processes relate to the role for the cells in cancer, rather than wider coverage of the basic biology of the cells.
  2. There is an inconsistent approach to discussing the different cell types that I found confusing in parts. The authors introduce the lineages and subsets by describing key differences between each of them, and compared to other T cells. However, the subsequent sections often discuss the cells as a collective as if they were uniform in function, despite the clear differences between the cells. This is most evidence in sections where characteristics of the cells (e.g., cytokine expression) are listed as a collective that suggests all of the cells share these features. However, in most instances, these were actually an accumulation of the different functions attributed to the lineages/subsets. My suggestion is that the characteristics of the lineages (and where appropriate, subsets) are described separately for different contexts, and then the similarities and differences between them as unconventional T cells are discussed.
  3. The role played by unconventional T cells and their potential as treatment targets could be more carefully discussed. The article seems to conclude that the importance of these cells in cancer is well established and there is good justification to now move to exploiting these cells in novel treatments. However, the link to cancer was mostly illustrated by providing a list of characteristics of the cells that had the potential to impact on the growth of cancer, or anti-tumour immunity, but without evidence of a definitive causal link. These associations and indirect findings are intriguing but do not yet represent compelling arguments for targeting the cells in new treatments. The article should include more discussion (or a separate section) detailing areas where more knowledge is required, including the need to determine how well the findings in mice apply to humans, and how to exploit the functions of some subsets, where others exist with apparently conflicting activities.

    This review of unconventional T cells is well researched and has the potential to be an excellent resource for researchers in this area, but changes should be made to improve it. The main areas that need attention are:

    1. The title of the article suggests a focus on the role of unconventional T cells in cancer, but a lot of detail is provided on the development and differentiation of these cells without substantial relevance to their role in cancer immunity. As such, the article sometimes reads like a broad review of these cells instead of being focused on cancer. I suggest those sections could be condensed so they are an introduction to the main themes, rather than equally weighted. The differentiation section is more obviously relevant, but I suggest the authors provide more specific discussion about how these processes relate to the role for the cells in cancer, rather than wider coverage of the basic biology of the cells.
    2. There is an inconsistent approach to discussing the different cell types that I found confusing in parts. The authors introduce the lineages and subsets by describing key differences between each of them, and compared to other T cells. However, the subsequent sections often discuss the cells as a collective as if they were uniform in function, despite the clear differences between the cells. This is most evidence in sections where characteristics of the cells (e.g., cytokine expression) are listed as a collective that suggests all of the cells share these features. However, in most instances, these were actually an accumulation of the different functions attributed to the lineages/subsets. My suggestion is that the characteristics of the lineages (and where appropriate, subsets) are described separately for different contexts, and then the similarities and differences between them as unconventional T cells are discussed. Some discussion as to whether distinct subsets are established during development or emerge only in response to external factors may be useful.
    3. The role played by unconventional T cells and their potential as treatment targets could be more carefully discussed. The article seems to conclude that the importance of these cells in cancer is well established and there is good justification to now move to exploiting these cells in novel treatments. However, the link to cancer was mostly illustrated by providing a list of characteristics of the cells that had the potential to impact on the growth of cancer, or anti-tumour immunity, but without evidence of a definitive causal link. These associations and indirect findings are intriguing but do not yet represent compelling arguments for targeting the cells in new treatments. The article should include more discussion (or a separate section) detailing areas where more knowledge is required, including the need to determine how well the findings in mice apply to humans, and how to exploit the functions of some subsets, where others exist with apparently conflicting activities.

Minor points:

  1. I recommend including references for the points listed in Table 1 .
  2. Consider including more information about how the cells are recruited to tumors, or where they are located if they mediate their effects from elsewhere. If elsewhere, how are they stimulated?

Author Response

First, we would like to thank the reviewer for her/his efforts in reviewing our manuscript as well as her/his positive comments and relevant suggestions. These have been addressed as follows:

Reviewer comments:

This review of unconventional T cells is well researched and has the potential to be an excellent resource for researchers in this area, but changes should be made to improve it. The main areas that need attention are:

  1. The title of the article suggests a focus on the role of unconventional T cells in cancer, but a lot of detail is provided on the development and differentiation of these cells without substantial relevance to their role in cancer immunity. As such, the article sometimes reads like a broad review of these cells instead of being focused on cancer. I suggest those sections could be condensed so they are an introduction to the main themes, rather than equally weighted. The differentiation section is more obviously relevant, but I suggest the authors provide more specific discussion about how these processes relate to the role for the cells in cancer, rather than wider coverage of the basic biology of the cells.

We thank the reviewer for this comment. Accordingly, we have focused the chapter 1 on aspects that were directly related to cancer immunology (functional development, functional versatility …). The general characteristics of UT cells have been condensed and referred to relevant existing reviews whenever possible. We hope that this edited section now conforms to the main theme of the review.

  1. There is an inconsistent approach to discussing the different cell types that I found confusing in parts. The authors introduce the lineages and subsets by describing key differences between each of them, and compared to other T cells. However, the subsequent sections often discuss the cells as a collective as if they were uniform in function, despite the clear differences between the cells. This is most evidence in sections where characteristics of the cells (e.g., cytokine expression) are listed as a collective that suggests all of the cells share these features. However, in most instances, these were actually an accumulation of the different functions attributed to the lineages/subsets. My suggestion is that the characteristics of the lineages (and where appropriate, subsets) are described separately for different contexts, and then the similarities and differences between them as unconventional T cells are discussed.

This is an important point raised by the reviewer. When drafting this review, we considered both approaches (separated or merged) to discuss UT cell fundamental biology. We chose the merged approach since it appeared to us more original than discussing lineages side by side. Please note that the other reviewer “like the merged approach”. However, we understand that it comes with some limitations especially for people that are not familiar to the field. Thus, we cautiously amended the chapter 1 to present the three subsets one by one whenever possible. We have also included a last paragraph to state the specific functions of each population.   

  1. The role played by unconventional T cells and their potential as treatment targets could be more carefully discussed. The article seems to conclude that the importance of these cells in cancer is well established and there is good justification to now move to exploiting these cells in novel treatments. However, the link to cancer was mostly illustrated by providing a list of characteristics of the cells that had the potential to impact on the growth of cancer, or anti-tumour immunity, but without evidence of a definitive causal link. These associations and indirect findings are intriguing but do not yet represent compelling arguments for targeting the cells in new treatments. The article should include more discussion (or a separate section) detailing areas where more knowledge is required, including the need to determine how well the findings in mice apply to humans, and how to exploit the functions of some subsets, where others exist with apparently conflicting activities.

We agree with the reviewer’s comment. Thus, we have included a section that discuss the areas that need to be further studied prior evaluation of the immunotherapeutic potential of these cells (l. 507-524). We thank the reviewer for this suggestion that improve the balance of our discussion.

Minor points:

  1. I recommend including references for the points listed in Table 1.

References have been now included in Table 1

  1. Consider including more information about how the cells are recruited to tumors, or where they are located if they mediate their effects from elsewhere. If elsewhere, how are they stimulated?

We thank the reviewer for this comment. In this review, we have focused on functions and phenotype associated with intratumoral UT cells (This is now clearly stated in the text (l. 228 and 400)). While the point of the reviewer is of interest, we believe that discussing how UT cells are recruited and/or can remotely control tumor growth is out of the scope of this review. Current evidence for a role of UT cells at distant sites are not compelling yet in the context of solid tumors.

Reviewer 2 Report

Barsac et al – Review on Protumoral Unconventional T cells in solid tumors

In the review by Barsac and colleague they provide an extensive overview of the pro-tumour roles for iNKT cells, gdT cells and MAIT cells. The review is very comprehensively referenced and covers the major aspects of UTs in cancer (some comments/suggestions are below). I like the merged approach although it can make it difficult to follow in the earlier background sections.

Major comments –

The merged approach of discussing the development of three cell types (iNKT, gdT & MAIT) made it complicated at points to follow – the authors should include a diagram outlining the development for clarity

The authors should include and discuss the following papers which highlight the role of the biome in MAIT cell development - 10.1126/science.aaw2719 & 10.1126/science.aax6624

Again the merged approach of discussing UT effector differentiation and release of cytokine has limitations (i.e type 2 MAITs & gdT cells).

Additionally the authors state (line131) that no bona fide UT subsets specialized in cytotoxicity exist – all UTs are equipped with fully cytotoxic arsenals and readily kill both in vitro and in vivo (as discussed in subsequent section) in particular gdT cells which are under investigation as immunotherapeutic agents based on both cytokine and killing capabilities. The authors should reconsider this statement

Figure 1 could be confusing for researchers new to UT cells e.g implies that all UTs respond to all cytokines and all can express FOX-P3 for example.

The single cell approach on iNKT cells (line 253-271) is far more accessible and clear.

Line 341 – the authors highlight the impact of the TME on iNKT cells. The authors should discuss the following paper - doi.org/10.1038/s41590-020-00848-3 & consider discussing the potential impact on MAIT cell metabolism (doi.org/10.3390/cancers13071582 & doi.org/10.4049/jimmunol.1801600)

Line 516 – authors provide single reference for UT based immunotherapies – should expand and include gd T cells and potentially discuss 10.1172/jci.insight.140074

In the conclusion the authors outline several potential therapeutic avenues – consider discussing CAR-UTs or targeting metabolism since it was a major component of review

Type setting issue – Starting on line 62 gamma delta symbols are corrupted

Author Response

First, we would like to thank the reviewer for her/his efforts in reviewing our manuscript as well as her/his positive comments and relevant suggestions. These have been addressed as follows:

Reviewer comments:

In the review by Barsac and colleague they provide an extensive overview of the pro-tumour roles for iNKT cells, gdT cells and MAIT cells. The review is very comprehensively referenced and covers the major aspects of UTs in cancer (some comments/suggestions are below). I like the merged approach although it can make it difficult to follow in the earlier background sections.

Major comments –

The merged approach of discussing the development of three cell types (iNKT, gdT & MAIT) made it complicated at points to follow – the authors should include a diagram outlining the development for clarity

Again the merged approach of discussing UT effector differentiation and release of cytokine has limitations (i.e type 2 MAITs & gdT cells).

The single cell approach on iNKT cells (line 253-271) is far more accessible and clear.

We thank the reviewer for these comments. This limitation has also been raised by the other referee. Thus, we now provide a revised version of chapter 1 in which the three lineages have been discussed one by one. We also included a last paragraph covering the differences between subsets. In parallel, we have condensed this first chapter to only keep meaningful information regarding the scope of the review. Finally, a new figure (Figure 1) outlining the functional development of UT cells is now provided as requested. We hope this new approach improves the clarity of this section.

The authors should include and discuss the following papers which highlight the role of the biome in MAIT cell development - 10.1126/science.aaw2719 & 10.1126/science.aax6624

Based on reviewer comments, the chapter 1 has been reorganised and shortened. Thus, the contribution of the microbiome in UT cell development is discussed in the section related to the microbiome 3.3. (l. 369).

Additionally the authors state (line131) that no bona fide UT subsets specialized in cytotoxicity exist – all UTs are equipped with fully cytotoxic arsenals and readily kill both in vitro and in vivo (as discussed in subsequent section) in particular gdT cells which are under investigation as immunotherapeutic agents based on both cytokine and killing capabilities. The authors should reconsider this statement

We apologize for any confusion regarding this point. We fully agree that all UT cell subsets are endowed with functional cytotoxic properties. However, we wanted to emphasize that, unlike conventional T cells and innate lymphoid cells (e.g. CD8+ CTL and NK cells), no UT cell subsets with specialized functions for cytotoxicity have been identified yet. Recent transcriptomic analysis on developing thymic iNKT and MAIT cells using single cell approach have revealed discrete cell clusters displaying gene signature of cells specialized in cytotoxicity (Legoux et al., 2019; Koay et al., 2019; Baranek et al., 2020; Krovy et al., 2020). However, the biological significance of these observations needs to be evaluated. We have rephrased our statement and moved it to the cytotoxic part (l. 176-179) to improve clarity. We thank the reviewer for pointing this out. 

Figure 1 could be confusing for researchers new to UT cells e.g implies that all UTs respond to all cytokines and all can express FOX-P3 for example.

We understand the reviewer’s point as we also considered depicting the various lineages and subsets in this figure. However, this strategy ended up with a messy figure with poor clarity. To avoid any possible confusion with the interpretation of this figure, we now stated in the figure legend that the regulatory processes pictured differ from one subset to another and cannot be considered as universal for all subsets. Regarding Foxp3 expression in UT cells, we replaced “Foxp3” and “PLZF” by the term “transcription factors”.

Line 341 – the authors highlight the impact of the TME on iNKT cells. The authors should discuss the following paper - doi.org/10.1038/s41590-020-00848-3 & consider discussing the potential impact on MAIT cell metabolism (doi.org/10.3390/cancers13071582 & doi.org/10.4049/jimmunol.1801600)

We thank the reviewer for these suggestions. Regarding the first manuscript (doi.org/10.1038/s41590-020-00848-3), it has already been cited in the paragraph regarding TME and γδT cells (l. 398-399) but no data on iNKT cells are provided in this paper. We have also included and discussed the two recent manuscripts from the Hogan Lab (l. 336-339).

Line 516 – authors provide single reference for UT based immunotherapies – should expand and include gd T cells and potentially discuss 10.1172/jci.insight.140074

We have included additional references on UT cell-based immunotherapies including for γδT cells (l. 527). The recent study on expansion of MAIT cells with enhanced cytotoxic functions has also been incorporated and discussed (l. 531).

In the conclusion the authors outline several potential therapeutic avenues – consider discussing CAR-UTs or targeting metabolism since it was a major component of review

We thank the reviewer for this suggestion. Even if it was already stated in the previous version of the manuscript, we have now dedicated several lines to the possibility of harnessing metabolism for UT cell function reprogramming (l. 553-557). As for the CAR-UT, although we acknowledge that this approach is promising, it seems to be out of the scope of the current review. Here, we wanted to focus on potential therapeutic strategies that aim at manipulating endogenous UT cells to block or “reverse” their protumoral functions.

Type setting issue – Starting on line 62 gamma delta symbols are corrupted

We thank the reviewer for pointing this out. We have now edited the text.

Round 2

Reviewer 1 Report

The authors have carefully addressed the recommendations of both Reviewers and the paper is now suitable for publication. The revised manuscript is well balanced and informative.